# Addressing Inverse Problems in Frame Restoration with Siamese Conditional Variational Autoencoders

## Abstract

Restoring missing information in video frames is a challenging inverse problem, particularly in applications such as autonomous driving and surveillance. This paper introduces the Siamese Masked Conditional Variational Autoencoder (SM-CVAE), a novel model that utilizes a Siamese network architecture with Siamese Vision Transformer (SiamViT) encoders. By leveraging the inherent similarities between paired frames, SMCVAE enhances the model's ability to accurately reconstruct missing content. This approach effectively tackles the problem of missing patches—often resulting from camera malfunctions—through advanced variational inference techniques. Experimental results demonstrate SMCVAE's superior performance in restoring lost information, highlighting its potential to solve complex inverse problems in real-world environments.

## 1 Introduction

The rapid advancement of imaging technologies has significantly broadened the applications of cameras, particularly in areas that demand high precision and reliability, such as autonomous driving systems (Geiger et al., 2012). In these systems, cameras serve as critical sensory components, providing real-time visual data essential for navigation and environmental perception (Muhammad et al., 2020; Agostinho et al., 2022). Accurate environmental perception is vital for ensuring both the safety and operational efficiency of autonomous vehicles. However, missing patches in video frames—caused by camera malfunctions, occlusions, or transmission errors—pose substantial challenges. These data gaps can severely undermine the perception systems of autonomous vehicles, leading to reduced reliability and overall performance, which may result in serious consequences in real-world scenarios (Mallozzi et al., 2019; Jebamikyous & Kashef, 2022). Despite extensive research aimed at addressing these issues, accurately restoring missing information in video frames remains a significant technical hurdle (Rota et al., 2023; Lamba & Mitra, 2022).

At the same time, video inpainting techniques have seen significant progress, offering methods for seamlessly editing video content. These techniques allow for the removal or modification of objects within video frames while preserving the temporal continuity of the footage, resulting in smooth and visually coherent sequences (Chang et al., 2019; Wu et al., 2023). However, video inpainting typically focuses on achieving aesthetic coherence rather than accurately restoring the original content. This distinction is crucial because, while inpainting produces visually plausible results, it often neglects the accurate recovery of missing information. In contrast, restoration tasks prioritize the authenticity of the content, aiming for precise reconstruction that restores video frames to their original state (Su et al., 2022).

To address the pressing need for reliable restoration methods, we propose the Siamese Masked Conditional Variational Autoencoder (SMCVAE), as illustrated in Figure 1. Given that adjacent video frames often share similar features and structures, the use of a Siamese network—with shared weights between two encoders—ensures consistent feature extraction from both masked and unmasked frames. This shared-weight approach allows both inputs to be represented in a common latent space, facilitating the model's ability to learn meaningful relationships between them and effectively reconstruct missing information. Moreover, weight sharing reduces the overall number of parameters, a critical consideration in deep learning applications where minimizing complexity

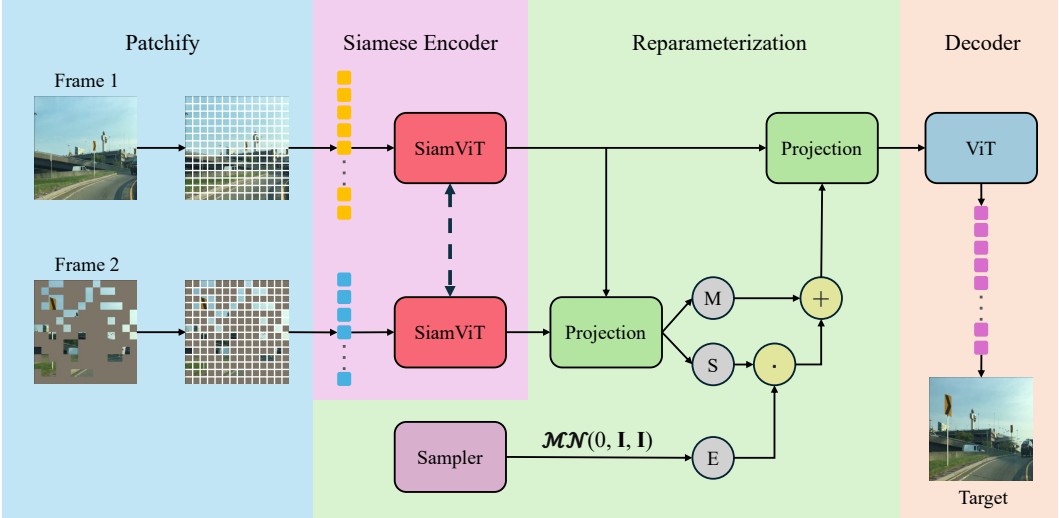

Figure 1: **SMCVAE Architecture.** SMCVAE takes a reference frame and a masked frame as inputs to restore the masked frame. By leveraging the inherent similarities between paired frames, this architecture effectively reconstructs the missing content.

without sacrificing performance is essential. By utilizing the same set of weights for encoding both masked and unmasked frames, the model maintains strong performance while being computationally more efficient—an important factor in frame restoration, where processing speed and memory usage are often limiting factors. Additionally, to handle uncertainty in corrupted frames and improve generalization, we incorporate variational inference, which further enhances the model's capabilities. In this paper, we evaluate this approach and demonstrate its advantages. The results show that SMCVAE is both efficient and accurate.

## 2 RELATED WORK

**Inverse problems.** Inverse problems, where one seeks to recover underlying causes from observed data, are central to various fields, including computer vision, medical imaging, and signal processing (Arridge et al., 2019; Hansen, 2010). These problems are typically formulated as reconstructing the original input $\mathbf{x}$ from an observation $\mathbf{y}$, where $\mathbf{y}$ is often corrupted by noise or incomplete data. Mathematically, the problem can be described as solving for $\mathbf{x}$ in the equation:

$$\mathbf{y} = \mathcal{A}(\mathbf{x}) + \mathbf{n}, \tag{1}$$

where $\mathcal{A}$ is a forward operator that maps the true signal $\mathbf{x}$ to the observation $\mathbf{y}$, and $\mathbf{n}$ represents noise or corruption. The goal is to invert this process and estimate $\mathbf{x}$ given $\mathbf{y}$. However, inverse problems are often ill-posed, meaning that small changes in $\mathbf{y}$ can lead to large variations in the estimated $\mathbf{x}$, or that a unique solution may not exist (Tarantola, 2005). Traditional approaches to solving inverse problems rely on regularization techniques, which introduce prior knowledge about the solution to stabilize the inversion (Bertero et al., 2021). For example, one widely used method is to minimize a regularized loss function:

$$\hat{\mathbf{x}} = \arg\min_{\mathbf{x}} \left( \|\mathcal{A}(\mathbf{x}) - \mathbf{y}\|^2 + \lambda \mathcal{R}(\mathbf{x}) \right), \tag{2}$$

where $\mathcal{R}(\mathbf{x})$ is a regularization term, and $\lambda$ controls the regularization strength. We leverage a similar deep learning-based approach to address the inverse problem of video frame restoration. Our proposed SMCVAE learns a latent representation of video frames, allowing it to accurately reconstruct missing or corrupted content. By incorporating variational inference and a Siamese architecture, SMCVAE provides a robust and efficient solution to this inverse problem (Bora et al., 2017).

**Siamese networks.** Siamese networks, characterized by their dual-branch architecture with shared weights, are highly effective for tasks involving similarity measurement and entity comparison. The

shared weights allow both branches to extract comparable features from different inputs, making Siamese networks well-suited for applications such as signature verification (Bromley et al., 1993), face verification (Chopra et al., 2005), and one-shot learning (Koch et al., 2015). Their versatility extends to contrastive learning, where they help in learning discriminative features by minimizing the distance between similar entities while maximizing the distance between dissimilar ones (Chen et al., 2020; He et al., 2020). This architecture has been instrumental in various computer vision tasks such as image classification, face recognition, and object tracking (Taigman et al., 2014; Bertinetto et al., 2016). Traditionally used in discriminative tasks, our research innovatively applies Siamese networks to generative modeling, opening up new possibilities in content restoration and generation.

**Variational inference.** Variational inference is a key technique in probabilistic modeling that approximates complex, often intractable, posterior distributions by optimizing an approximate distribution, $q(\mathbf{z})$, to match the true posterior, $p(\mathbf{z}|\mathbf{x})$, using the Kullback-Leibler (KL) divergence as a measure of information loss (Blei et al., 2017; Jordan et al., 1999). This approach balances computational efficiency with model expressiveness, making it particularly useful in scalable machine learning applications. By minimizing the KL divergence, variational inference enables efficient posterior estimation in large datasets and complex models such as Bayesian neural networks (Blundell et al., 2015). Its role is central to the development of models like the Variational Autoencoder (VAE) (Kingma, 2013; Rezende et al., 2014), which leverages variational inference to learn latent variable models in a generative framework. The Conditional Variational Autoencoder (CVAE) extends this framework to handle conditional dependencies, making it well-suited for tasks such as video frame restoration, where generative models need to account for auxiliary variables (Sohn et al., 2015). Variational inference not only enhances computational tractability but also allows for more flexible and expressive modeling, underscoring its importance in both generative modeling and Bayesian deep learning (Hoffman et al., 2013; Tucker et al., 2017).

## 3 METHOD

**Siamese encoder.** The encoding process begins by converting each pair of video frames into a structured sequence of patches. Specifically, frames $\mathbf{A}_1$ and $\mathbf{A}_2 \in \mathbb{R}^{H \times W \times C}$ are transformed into sequences of 2D flattened patches, denoted as $\mathbf{X}_1$ and $\mathbf{X}_2 \in \mathbb{R}^{N \times (P^2 \cdot C)}$. In this notation, $H \times W$ represents the dimensions of the original frames, $C$ is the number of channels, $P \times P$ defines the resolution of each patch, and $N = \dfrac{HW}{P^2}$ represents the total number of patches extracted from each frame. To accommodate the paired frame structure, our method utilizes the SiamViT architecture, which leverages Siamese Vision Transformers (SiamViT) with shared weights for efficient processing (Dosovitskiy et al., 2020). This architecture processes paired frames consisting of one unaltered frame and one masked frame, enabling the model to handle the unique challenges of video frame restoration by focusing on the similarities and discrepancies between the two frames.

The SiamViT architecture employs a series of alternating Multiheaded Self-Attention (MSA) and Multilayer Perceptron (MLP), a design inspired by the ViT model (Vaswani et al., 2017; Dosovitskiy et al., 2020). Each block is preceded by Layer Normalization (LN) to stabilize training, while residual connections are incorporated after each block to facilitate gradient flow during backpropagation, which is crucial for efficient learning in deep networks (He et al., 2016). These components work together to maintain a stable and efficient flow of information through the network, enhancing the model's ability to learn robust feature representations from the video data.

The SiamViT operates according to the following mathematical formulation:

$$\mathbf{Y}_{i,0} = (\mathbf{W}_e \mathbf{X}_i^\top + \mathbf{B}_e)^\top + \mathbf{P}_e, \tag{3}$$

$$\mathbf{Y}'_{i,l} = \text{MSA}_l(\text{LN}(\mathbf{Y}_{i,l-1})) + \mathbf{Y}_{i,l-1}, \tag{4}$$

$$\mathbf{Y}_{i,l} = \text{MLP}_l(\text{LN}(\mathbf{Y}'_{i,l-1})) + \mathbf{Y}'_{i,l-1}, \tag{5}$$

$$\mathbf{U}_i = (\mathbf{W}_u \text{LN}(\mathbf{Y}_{i,L})^\top + \mathbf{B}_u)^\top, \tag{6}$$

where $\mathbf{W}_e$ and $\mathbf{B}_e$ are the weights and biases used for embedding the patches, $\mathbf{P}_e$ represents the positional embeddings, and $\mathbf{W}_u$ and $\mathbf{B}_u$ are the final projection weights and biases.

Following this encoding step, the trainable mask token $\mathbf{t}^\top$ is replicated according to the cardinality of the set $\mathcal{P}$, which denotes the indices of the masked patches. This replication results in a matrix

ready for integration with $\mathbf{U}_2$, the encoded representation of the masked frame. The merging of $\mathbf{U}_1$ (from the unaltered frame) and $\mathbf{U}_2$, along with the masked tokens, is performed as follows:

$$\mathbf{T} = \text{Repeat}(\mathbf{t}^\mathsf{T}, |\mathcal{P}|), \tag{7}$$

$$\mathbf{U} = [\mathbf{U}_1, \text{Convert}(\mathbf{U}_2, \mathcal{P}, N+1) + \text{Convert}(\mathbf{T}, \mathcal{P}^c, N+1)], \tag{8}$$

where $\mathcal{P}$ signifies the indices of the masked patches, $\mathcal{P}^c$ represents the complement of $\mathcal{P}$, $[\cdot, \cdot]$ denotes the concatenation operation, and $|\cdot|$ represents the cardinality of the set. This approach ensures that the model can focus on recovering the missing content by leveraging both the unaltered frame and the masked frame, thus improving its restoration capabilities (Bao et al., 2021; He et al., 2022).

**Reparameterization.** After the siamese encoder processes the input, the extracted features pass through the reparameterization layer, where they are mapped into a Gaussian-distributed latent space. This step is crucial for enhancing the model's ability to generate diverse and nuanced representations by enabling stochasticity in the latent variables (Kingma, 2013; Rezende et al., 2014). The reparameterization trick allows for efficient backpropagation through stochastic layers, a key innovation in variational autoencoders (VAEs) that facilitates end-to-end training. Mathematically, the operation is defined as follows:

$$\mathbf{M} = (\mathbf{W}_{\mathrm{m}}\mathbf{U}^\mathsf{T} + \mathbf{B}_{\mathrm{m}})^\mathsf{T}, \tag{9}$$

$$\mathbf{S} = (\mathbf{W}_{\mathrm{s}}\mathbf{U}^\mathsf{T} + \mathbf{B}_{\mathrm{s}})^\mathsf{T}, \tag{10}$$

$$\mathbf{Z} = \mathbf{M} + \mathbf{S} \odot \mathbf{E}, \tag{11}$$

where $\mathbf{W}_{\mathrm{m}}$ and $\mathbf{W}_{\mathrm{s}}$ are weight matrices, and $\mathbf{B}_{\mathrm{m}}$ and $\mathbf{B}_{\mathrm{s}}$ are bias vectors, corresponding to the mean ($\mathbf{M}$) and standard deviation ($\mathbf{S}$) of the latent space, respectively. The matrix $\mathbf{E}$ is sampled from a multivariate normal distribution $\mathcal{MN}_{(N+1) \times D'}(\mathbf{0}, \mathbf{I}, \mathbf{I})$, introducing a stochastic element through the Hadamard product $\odot$. This process allows the model to efficiently sample from the latent space while maintaining differentiability, a critical aspect of variational inference in deep generative models (Kingma et al., 2019). The reparameterization trick thus plays an essential role in generating continuous and diverse latent representations, key for the model's generative capabilities.

**Decoder.** The decoder is designed as a specialized Vision Transformer (ViT) (Dosovitskiy et al., 2020), which plays a crucial role in reconstructing the original visual content by predicting the appearance of individual patches in pixel space. This transformation takes the latent representations and converts them back into the visual domain, with the objective of filling in missing content with high fidelity and accuracy (Esser et al., 2021; Parmar et al., 2018). The decoder's architecture allows for precise reconstruction by leveraging self-attention mechanisms to capture the global context of the image (Vaswani et al., 2017). The mathematical operations that govern this reconstruction process are as follows:

$$\mathbf{V}_0 = (\mathbf{W}_{\mathrm{d}}[\mathbf{Z}, \mathbf{U}_1]^\mathsf{T} + \mathbf{B}_{\mathrm{d}})^\mathsf{T} + \mathbf{P}_{\mathrm{d}}, \tag{12}$$

$$\mathbf{V}_l' = \text{MSA}_l'(\text{LN}(\mathbf{V}_{l-1})) + \mathbf{V}_{l-1}, \tag{13}$$

$$\mathbf{V}_l = \text{MLP}_l'(\text{LN}(\mathbf{V}_{l-1}')) + \mathbf{V}_{l-1}', \tag{14}$$

$$\mathbf{O} = (\mathbf{W}_{\mathrm{o}}\text{LN}(\mathbf{V}_{L'})^\mathsf{T} + \mathbf{B}_{\mathrm{o}})^\mathsf{T}, \tag{15}$$

where $\mathbf{W}_{\mathrm{d}}$ and $\mathbf{B}_{\mathrm{d}}$ represent the weights and biases of the decoder's embedding layer, $\mathbf{P}_{\mathrm{d}}$ specifies the positional embeddings within the decoder, and $\mathbf{W}_{\mathrm{o}}$ and $\mathbf{B}_{\mathrm{o}}$ denote the weights and biases of the output layer, respectively. These operations leverage the ViT's MSA and MLP modules to iteratively refine the latent representations and generate a coherent output in pixel space. The decoder's ability to accurately reconstruct missing patches is crucial for high-quality video frame restoration (Chen et al., 2021), ensuring that the synthesized content closely matches the original.

To integrate the predicted content with the original unmasked patches, the following operation is employed:

$$\mathbf{R} = \text{Convert}([\mathbf{0}^\mathsf{T}; \mathbf{X}_2], \mathcal{P}, N) + \text{Convert}(\mathbf{O}, \mathcal{P}^c, N), \tag{16}$$

where $[\cdot\,;\cdot]$ signifies vertical concatenation. This operation allows for the predicted patches to be merged with the original content, ensuring that the final reconstruction, $\mathbf{R}$, retains both the predicted and original information in their correct spatial positions.

**Loss function.** Inspired by the $\beta$-VAE framework (Higgins et al., 2017), we adopt an isotropic unit Gaussian $\mathcal{MN}(\mathbf{0}, \mathbf{I}, \mathbf{I})$ as the prior distribution. This choice leads to a constrained optimization problem that maximizes the expected log-likelihood of the ground truth $\mathbf{G}$ given the latent representations $\mathbf{Z}$, while constraining the Kullback-Leibler (KL) divergence to ensure a controlled information bottleneck. The optimization problem is formally expressed as:

$$\min_{\phi,\theta} \; -\mathbb{E}_{\mathbf{X}_1,\mathbf{X}_2\sim\mathcal{D}} \left[ \mathbb{E}_{q_\phi(\mathbf{Z}|\mathbf{X}_1,\mathbf{X}_2)} \log p_\theta(\mathbf{G} \mid \mathbf{Z}) \right],$$
$$\text{s.t. } D_{\text{KL}} \left( q_\phi(\mathbf{Z} \mid \mathbf{X}_1, \mathbf{X}_2) \| p(\mathbf{Z}) \right) \leq \epsilon, \tag{17}$$

where $\phi$ and $\theta$ represent the parameters of the encoder and decoder, respectively. The input pairs $\mathbf{X}_1$ and $\mathbf{X}_2$ come from the dataset $\mathcal{D}$, with $q\phi$ denoting the variational posterior and $p\theta$ the likelihood of the reconstruction given the latent space. The KL divergence term $D_{\text{KL}}$ ensures regularization by constraining the learned latent distribution to the prior $p(\mathbf{Z})$, with $\epsilon$ controlling the strength of the regularization. This formulation balances reconstruction fidelity and latent space regularization, a key feature of variational autoencoders (Kingma, 2013; Rezende et al., 2014).

To further refine the model, we reconceptualize the loss function within a Lagrangian framework:

$$\mathcal{F}(\theta, \phi, \beta) = -\mathbb{E}_{q_\phi(\mathbf{Z}|\mathbf{X}_1,\mathbf{X}_2)} [\log p_\theta(\mathbf{G} \mid \mathbf{Z})] + \beta \left( D_{\text{KL}} \left( q_\phi(\mathbf{Z} \mid \mathbf{X}_1, \mathbf{X}_2) \| p(\mathbf{Z}) \right) - \epsilon \right), \tag{18}$$

where $\mathcal{F}$ denotes the Lagrangian, incorporating the expected log-likelihood of the ground truth and the KL divergence, which is scaled by $\beta$ to control the trade-off between reconstruction quality and adherence to the prior distribution (Burgess et al., 2018).

Since $\epsilon$ is fixed and does not influence the optimization directly, it can be omitted from operational calculations. The overall loss function integrates both the reconstruction loss ($\mathcal{L}_\text{r}$) and the KL divergence loss ($\mathcal{L}_\text{KL}$), as follows:

$$\mathcal{L} = \mathcal{L}_\text{r} + \beta \cdot \mathcal{L}_\text{KL}, \tag{19}$$

where $\beta$ regulates the balance between these two components.

The reconstruction loss, $\mathcal{L}_\text{r}$, measures the difference between the original data and the reconstructed data, which is crucial for evaluating restoration accuracy. It is defined as:

$$\mathcal{L}_\text{r} = \frac{1}{P^2 C |\mathcal{P}|} \|\mathbf{R} - \mathbf{G}\|_\text{F}^2, \tag{20}$$

where $\| \cdot \|_\text{F}$ is the Frobenius norm, capturing the squared sum of pixel-wise differences across all channels.

The KL divergence loss $\mathcal{L}_\text{KL}$, which quantifies the dissimilarity between the learned latent distribution and the prior, is given by:

$$\mathcal{L}_\text{KL} = \frac{\|\mathbf{M}\|_\text{F}^2 + \|\mathbf{S}\|_\text{F}^2 - \sum_{i=1}^{N+1} \sum_{j=1}^{D'} \log \mathbf{S}_{ij}}{2(N+1)D'} - \frac{1}{2}. \tag{21}$$

To optimize the model, we aim to minimize the overall loss:

$$\begin{aligned}
\phi^*, \theta^* &= \arg\min_{\phi,\theta} \mathcal{F}(\theta, \phi, \beta) \\
&= \arg\min_{\phi,\theta} (-\mathbb{E}_{q_\phi(\mathbf{Z}|\mathbf{X}_1,\mathbf{X}_2)} \log p_\theta(\mathbf{G} \mid \mathbf{Z}) + \beta \cdot D_{\text{KL}}(q_\phi(\mathbf{Z} \mid \mathbf{X}_1, \mathbf{X}_2) \| p(\mathbf{Z}))) \\
&= \arg\min_{\phi,\theta} (\mathcal{L}_\text{r} + \beta \cdot \mathcal{L}_\text{KL}) \\
&= \arg\min_{\phi,\theta} \mathcal{L}.
\end{aligned} \tag{22}$$

This ensures that the model achieves high-quality reconstructions while maintaining a regularized and meaningful latent space, balancing the trade-off between reconstruction and latent space regularization (Alemi et al., 2018).

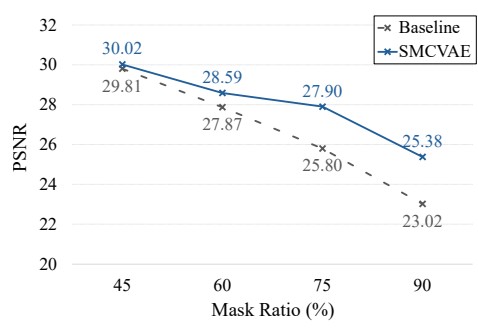 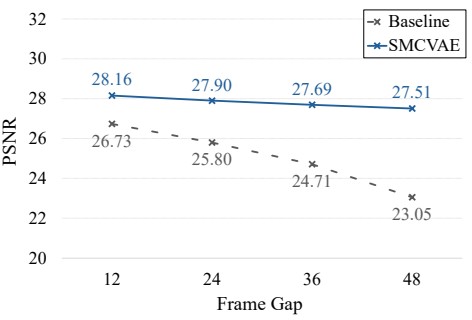

Figure 2: **Mask ratio.** Our SMCVAE exhibits remarkable resilience, maintaining high restoration quality over a spectrum of mask ratios.

Figure 3: **Frame gap.** Our SMCVAE maintains impressive stability in handling varying frame gaps with minimal performance impact.

## 4 EXPERIMENTS

### 4.1 EXPERIMENT SETUP

**Dataset.** For our experiments, we utilize the BDD100K dataset, a comprehensive collection of images and videos captured across various driving conditions and environmental scenarios (Yu et al., 2020). To streamline the training process and reduce computational overhead, we select a representative subset of the dataset.

**Masking strategy.** To replicate real-world scenarios of data loss or image corruption, we implement a strategic masking procedure. In each image pair, one frame is selectively occluded to simulate partial data loss or corruption, as commonly observed in dynamic video feeds due to transmission errors, occlusions, or sensor malfunctions (Pathak et al., 2016; Zhao et al., 2016). The unmasked frame serves as a reference for restoration, enabling our model to learn how to effectively reconstruct missing or corrupted content.

**Baseline.** Our baseline model consists of two independent ViT encoders, each tasked with separately processing the masked and unmasked input frames, followed by a ViT decoder that reconstructs the corrupted frames. This setup provides a strong foundation for comparison, allowing us to measure the improvements brought by our proposed model.

### 4.2 MODEL ROBUSTNESS

To assess the resilience and adaptability of the SMCVAE model, we conducted simulations that reflect challenges commonly encountered in real-world scenarios. These tests are designed to evaluate the model's robustness in handling data loss and temporal discontinuities, both of which are prevalent in practical applications.

**Mask ratio.** To evaluate the model's robustness under varying levels of data loss, we tested its performance across a range of mask ratios. This analysis provided valuable insights into the model's ability to handle different degrees of data degradation. The results, illustrated in Figure 3, demonstrate SMCVAE's strong ability to reconstruct video frames even under severe masking conditions. This highlights the model's reliability and effectiveness across a wide range of data corruption severities.

**Frame gap.** We also explored the model's performance under varying frame gaps, which simulate temporal discontinuities. As shown in Figure 3, SMCVAE consistently maintains stable performance across different frame gaps, underscoring its adaptability. This consistent performance demonstrates the model's robustness in restoring accurate content, regardless of the temporal distance between frames.

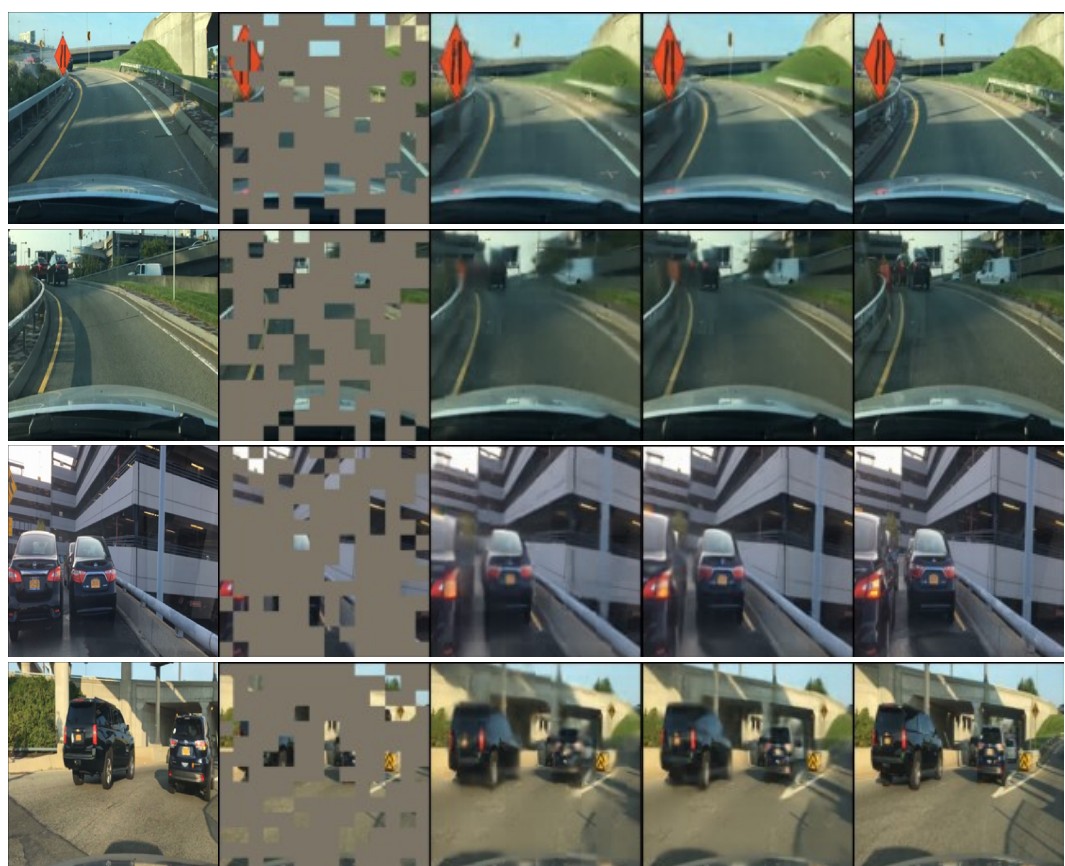

Figure 4: **Comparative visualization.** Our SMCVAE model's performance in reconstructing occluded regions is visually compared with other methods, showcasing its superior ability to restore image fidelity even in heavily masked scenarios.

### 4.3 QUALITATIVE ANALYSIS

To further assess the SMCVAE model's capabilities, we conducted a qualitative analysis focusing on the visual quality of the restoration outputs. The visual results, as illustrated in Figure 4, offer a direct comparative view of SMCVAE's performance against other leading models in the field. The visual results not only underscore SMCVAE's superior restoration capabilities but also highlight its effectiveness in producing visually coherent and detailed images, further affirming its excellence in the domain of video frame restoration.

### 4.4 ABLATION STUDIES

To evaluate the impact of key design choices in SMCVAE, we performed a series of ablation studies.

**Reparameterization.** We examined the role of reparameterization within the SMCVAE architecture by comparing model performance with and without this feature. As shown in Table 1a, We compare the result about no reparameterization, variational inference and conditional variational inference. This result highlights the essential role of reparameterization in enhancing the model's overall effectiveness, establishing it as a vital component for superior video frame restoration.

**Lagrange multiplier.** A crucial aspect of configuring the SMCVAE model is tuning the Lagrange multiplier ($\beta$), which balances the model's regularization strength and restoration performance. As illustrated in Table 1b, we explored the model's sensitivity to different $\beta$ values. The analysis demonstrates how varying levels of regularization affect the model's capacity to accurately reconstruct

Table 1: **Ablation studies.** We conducted a series of ablation experiments to evaluate the impact of key components in SMCVAE. Default settings are highlighted in `gray`.

(a) **Reparameterization.** Incorporating reparameterization improves performance.

| Reparam | PSNR |
|---------|-------|
| None | 20.55 |
| Var | 23.27 |
| Cond | **23.40** |

(b) **Lagrange multiplier.** The results show the model's sensitivity to different $\beta$ values.

| $\beta$ | PSNR |
|---------|-------|
| $10^0$ | 21.97 |
| $10^{-1}$ | 22.36 |
| $10^{-2}$ | **23.40** |
| $10^{-3}$ | 23.21 |

(c) **Encoder design.** The Siamese encoder performs similarly to the dual encoder.

| Type | PSNR |
|---------|-------|
| Dual | **23.42** |
| Siamese | 23.40 |

video frames. The findings underscore the importance of fine-tuning $\beta$, with $\beta = 10^{-2}$ identified as the optimal value for maximizing performance across key evaluation metrics.

**Encoder design.** Implementing a Siamese encoder architecture, which utilizes weight sharing between the two encoders, offers the significant advantage of reducing the model's total parameter count by approximately 43%. As demonstrated in Table 1c, the Siamese architecture achieves performance comparable to that of a dual-encoder configuration without weight sharing. This result demonstrates that weight sharing effectively preserves the model's restoration capabilities while substantially decreasing computational complexity and memory usage.

## 5 DISCUSSION

The results of our experiments demonstrate the effectiveness of our SMCVAE in handling complex video frame restoration tasks. By incorporating a Siamese architecture with shared weights, the model not only reduces the number of parameters but also maintains high reconstruction quality, even under severe data loss conditions. This aligns with previous findings on the benefits of weight-sharing in Siamese networks for reducing model complexity while preserving performance (Koch et al., 2015; Bromley et al., 1993). Our ablation studies further highlight the importance of key architectural choices, such as the use of reparameterization and the fine-tuning of the Lagrange multiplier, both of which significantly enhance the model's performance. These findings are consistent with existing literature on $\beta$-VAE, where tuning $\beta$ improves the balance between reconstruction fidelity and latent space regularization (Higgins et al., 2017).

In comparison with the baseline model, which utilizes dual independent ViT encoders, SMCVAE consistently outperforms in terms of reconstruction accuracy and robustness to varying degrees of corruption. The model's ability to preserve essential content while filling in missing details with high fidelity demonstrates its potential for real-world applications, particularly in domains like autonomous driving, where data integrity is critical (Yu et al., 2020; Geiger et al., 2012). This performance, especially in scenarios involving severe masking or large frame gaps, underscores the robustness of the SMCVAE architecture, which is capable of handling complex temporal dependencies in video data.

Despite the promising results, there remain challenges, such as further optimizing the model's efficiency for real-time applications and exploring its performance in more diverse and extreme conditions. Future work could focus on extending the model's capabilities to multi-modal datasets or integrating it with state-of-the-art video restoration techniques to further enhance its generalizability and performance.

## 6 CONCLUSION

In this paper, we introduced the Siamese Masked Conditional Variational Autoencoder (SMCVAE), a novel architecture tailored for video frame restoration in the presence of partial data loss or corruption. Leveraging a Siamese encoder with weight sharing, SMCVAE strikes an effective balance between model complexity and performance. The integration of a conditional variational frame-

work further enhances the model's ability to manage high levels of uncertainty in corrupted frames. Our experimental results show that SMCVAE consistently outperforms baseline models, particularly in challenging scenarios involving severe masking or temporal discontinuities, demonstrating its robustness and superior reconstruction quality.

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
