# OpenReview forum: "Addressing Inverse Problems in Frame Restoration with Siamese Conditional Variational Autoencoders"
_ICLR.cc/2025/Conference — ICLR 2025 Conference Withdrawn Submission_

### Official Review · Reviewer_HYNH · 2024-10-31

**Soundness:** 1
**Presentation:** 1
**Contribution:** 1
**Rating:** 3
**Confidence:** 5

**Summary:**

This paper introduces the Siamese Masked Conditional Variational Autoencoder to solve the inverse problem in frame restoration. It is typically a technical application report rather than a research paper. Experiments on BDD100K datasets demonstrated the practicality of the applied models. No clear novelty or contribution can be found in this work.

**Strengths:**

The experiment settings seem reasonable and considerable to evaluate the effectiveness of the proposed work.

**Weaknesses:**

- Lack novelty and contribution. This work simply applied the well-known SiamViT architecture to solve the inverse problem in frame restorations, without any observable technical improvements or promising designs. Most parts of the method section only introduce the details and training loss of SiamViT.
- Poor presentation. Many figures are less informative and confusing to readers. For example, Figure 2 lacks the details of some components in the proposed framework. The names of comparison methods are missing in Figure 4.
- Insufficient experiments. This work only evaluated a single task in inverse problems with a dataset. Besides, more comparison methods and quantitative results should be included.

**Questions:**

Please CAREFULLY revise the paper if the authors plan to submit it to the next conference.

---

### Official Review · Reviewer_1jYM · 2024-11-02

**Soundness:** 1
**Presentation:** 2
**Contribution:** 2
**Rating:** 3
**Confidence:** 5

**Summary:**

To recover missing information from corrupted video frames, this paper introduces SMCVAE, a Siamese network with Siamese Vision Transformer (SiamViT) encoders. Specifically, SMCVAE leverages complementary information from paired frames to reconstruct missing contents. Experimental results indicate that SMCVAE effectively restores missing regions in corrupted video frames.

**Strengths:**

1. This paper combines Siamese networks and conditional variational inference for frame restoration. The use of SiamViT with shared weights reduces the parameters and improves computational efficiency.
1. The application of video frame restoration in areas like autonomous driving and surveillance is highly relevant and addresses practical needs.

**Weaknesses:**

1. The paper lacks comprehensive quantitative comparisons. Including experiments against state-of-the-art video inpainting or generative restoration methods would provide a more robust evaluation and highlight the method’s advantages.
2. The paper does not adequately explain how the **Siamese Encoder** enables effective video frame restoration. A shared-weight encoder does not directly leverage complementary information from paired frames for information completion.
3. The role of **variational inference** in video frame restoration is not clearly justified. Although the authors state that variational inference addresses “uncertainty” in corrupted frames, the nature of this uncertainty and how variational inference mitigates it is unclear. A more thorough explanation is needed to support this claim.

**Questions:**

1. Has the computational efficiency of SMCVAE been tested in real-time or near-real-time scenarios, especially given the relevance to autonomous driving applications?
2. The paper evaluates SMCVAE on the BDD100K dataset, which is specific to driving environments. How would the model perform on other types of datasets with different visual characteristics, such as indoor scenes or natural landscapes?
3. What is the motivation behind choosing SiamViT over other potential architectures for the encoder?
4. Does the paper identify any specific limitations of SMCVAE when applied to different frame corruption patterns or motion dynamics?

---

### Official Review · Reviewer_VS1M · 2024-11-04

**Soundness:** 3
**Presentation:** 3
**Contribution:** 1
**Rating:** 3
**Confidence:** 3

**Summary:**

This paper introduced Siamese masked conditional variational autoencoder for video frame restoration aiming to obtain more accurate image restoration result. In particular, the proposed method mainly adopts a VAE with Siamese encoder. The proposed method is evaluated on BDD100K dataset and shows promising results.

**Strengths:**

The paper tackles an important problem of image restoration for video sequences. Moreover, the Siamese encoder based VAE seems effective in handling the task.

**Weaknesses:**

-	Limited Novelty.  While the paper introduced Siamese encoder-based VAE for image restoration, each module is not new. Therefore there is limited novelty.
-	The proposed method didn’t compare with any existing and published video frame restoration method. It is hard to evaluate how good the method is compared with existing methods in tackling the video frame restoration task.
-	It is really not clear whether the proposed method is necessary because we can leverage information from dense frames in the video sequence and get pretty good results.
Given lack of novelties and comparisons with existing method, the reviewer would recommend Reject.

**Questions:**

Due to lack of comparison with any existing video restoration method, it is really difficult to evaluate the performance compared with existing methods. The authors are encouraged to compare with relevant works in the field to demonstrate their performance.

---

### Official Review · Reviewer_eEBH · 2024-11-05

**Soundness:** 2
**Presentation:** 2
**Contribution:** 2
**Rating:** 3
**Confidence:** 5

**Summary:**

The paper introduces Siamese Masked Conditional Variational Autoencoder (SMCVAE) for restoring missing information in video frames. It employs a Siamese network architecture with Siamese Vision Transformer encoders to leverage the similarities between paired frames, and incorporates variational inference to address uncertainty in corrupted frames. Evaluated on the BDD100K dataset, the SMCVAE demonstrates robustness across various mask ratios and frame gaps.

**Strengths:**

1. The integration of variational inference techniques allows the model to handle uncertainty in corrupted frames.
2. SMCVAE demonstrates resilience across a range of mask ratios and frame gaps, indicating its ability on varying degrees of data loss and temporal discontinuities.

**Weaknesses:**

1. The primary weakness is the narrow scope of comparative analysis, with the SMCVAE being benchmarked against only a single baseline model. This might not adequately capture its performance relative to a broader spectrum of state-of-the-art methods, potentially limiting the assessment of its true capabilities in video frame restoration.
2. The experiments are only centered on the BDD100K dataset. A broader dataset could provide insights into the model's generalizability and robustness in varied real-world scenarios.
3. This work's innovation is somewhat limited, primarily offering a novel combination of existing techniques, including frame pairs, variational inference, siamese architecture, and SiamViT encoders, rather than introducing transformative new methodologies in the field of video frame restoration.

**Questions:**

Please see the above Weaknesses.

---

### Official Review · Reviewer_i81i · 2024-11-05

**Soundness:** 3
**Presentation:** 2
**Contribution:** 1
**Rating:** 3
**Confidence:** 4

**Summary:**

This paper proposes a Siamese Masked Conditional Variational Autoencoder for restoring missing information in video frames. SMCVAE uses a Siamese network architecture with SiamViT encoders and incorporates variational inference to learn a latent representation of video frames and reconstruct missing or corrupted content. The authors demonstrate their approach through a series of experiments and ablation studies on a single dataset - BDD100K.

**Strengths:**

- The general concept of using siamese learning + variational inference is interesting and worth exploring. If the claims of robustness and performance hold up under more rigorous evaluation, this could be a very useful approach.

- The paper is well-structured and written clearly (although the figures and tables are poorly presented as noted below). It effectively explains the problem of learning a good represented, the proposed solution, and the results.

**Weaknesses:**

- While the specific combination of techniques in SMCVAE may be new, the fundamental idea of using masked autoencoders for image and video restoration is not. Several existing works already employ masked autoencoding within Vision Transformer architectures for similar tasks. The authors should more clearly articulate the specific novel aspects of their approach beyond the general combination of techniques.

- The experiments, as presented, raise several concerns. The lack of details about the "representative subset" chosen from the BDD100K dataset makes it difficult to evaluate the generalizability of the results. More information about the specific masking strategy implemented, including the size, shape, and distribution of masked regions, is necessary to understand its realism and relevance to real-world scenarios. Including stronger baseline comparisons with other state-of-the-art video frame restoration techniques, particularly those employing masked autoencoding or similar approaches (eg. V-JEPA).

- The presentation of figures and tables needs improvement. The aspect ratio of all the examples in Figure 4 are incorrect. Figure 4 should also include a visual comparison to the baseline and existing methods.

**Questions:**

N/A

---

### Note · Authors · 2025-01-21

I have read and agree with the venue's withdrawal policy on behalf of myself and my co-authors.